# Delving into the Mechanisms of Sponge-Associated *Enterobacter* against Staphylococcal Biofilms

**DOI:** 10.3390/molecules28124843

**Published:** 2023-06-19

**Authors:** Anna Luiza Bauer Canellas, Bruno Francesco Rodrigues de Oliveira, Suzanne de Oliveira Nunes, Camila Adão Malafaia, Ana Claudia F. Amaral, Daniel Luiz Reis Simas, Ivana Correa Ramos Leal, Marinella Silva Laport

**Affiliations:** 1Instituto de Microbiologia Paulo de Góes, Universidade Federal do Rio de Janeiro, Rio de Janeiro 21941-590, Brazil; annaluiza@micro.ufrj.br (A.L.B.C.); bfroliveira@id.uff.br (B.F.R.d.O.); suzannenunes84@gmail.com (S.d.O.N.); 2Departamento de Microbiologia e Parasitologia, Instituto Biomédico, Universidade Federal Fluminense, Niterói 24210-130, Brazil; 3Laboratório de Produtos Naturais e Ensaios Biológicos, DPNA, Faculdade de Farmácia, Universidade Federal do Rio de Janeiro, Rio de Janeiro 21941-902, Brazil; camilar.adao@gmail.com (C.A.M.); danielsimas16@gmail.com (D.L.R.S.); 4Laboratório de Plantas Medicinais e Derivados, Farmanguinhos, Fiocruz, Rio de Janeiro 21041-250, Brazil; ana.amaral@fiocruz.br; 5Bio Assets Biotecnologia, São Paulo 05511-010, Brazil

**Keywords:** antibiofilm, biofilm infections, *Enterobacter*, Porifera, *Staphylococcus aureus*

## Abstract

Staphylococci are one of the most common causes of biofilm-related infections. Such infections are hard to treat with conventional antimicrobials, which often lead to bacterial resistance, thus being associated with higher mortality rates while imposing a heavy economic burden on the healthcare system. Investigating antibiofilm strategies is an area of interest in the fight against biofilm-associated infections. Previously, a cell-free supernatant from marine-sponge-associated *Enterobacter* sp. inhibited staphylococcal biofilm formation and dissociated the mature biofilm. This study aimed to identify the chemical components responsible for the antibiofilm activity of *Enterobacter* sp. Scanning electron microscopy confirmed that the aqueous extract at the concentration of 32 μg/mL could dissociate the mature biofilm. Liquid chromatography coupled with high-resolution mass spectrometry revealed seven potential compounds in the aqueous extract, including alkaloids, macrolides, steroids, and triterpenes. This study also suggests a possible mode of action on staphylococcal biofilms and supports the potential of sponge-derived *Enterobacter* as a source of antibiofilm compounds.

## 1. Introduction

Biofilm-associated infections are a significant concern in medical settings, with evidence suggesting they are involved in approximately 75% of human microbial infections, particularly when medical devices such as catheters and orthopedic implants are involved. This significantly increases mortality rates and imposes an economic burden on healthcare systems [1]. Conventional approaches using antimicrobials may not be efficient in clearing chronic biofilm infections and, in many cases, promote the emergence of resistance, which is usually observed for staphylococci, one of the most common causes of biofilm-related infections [2]. Currently, the most efficient means of eradicating biofilms remains the surgical removal of the infected device or debridement of the wound or bone, but this may lead to other complications [3]. Thus, new approaches are urgently needed to prevent and inhibit bacterial adhesion and biofilm formation mechanisms [4].

Antibiofilm strategies are mainly of two types: those involving the inhibition or prevention of new biofilm formation and those based on the dispersal or eradication of existing biofilms [5]. For instance, the interest in describing the antibiofilm activity of quorum-quenching molecules, matrix-degrading enzymes, antimicrobial peptides, (bio)surfactants, fatty acids, and many other substances has peaked in recent decades [4,6,7,8,9]. New approaches have been proposed. Molecular docking studies were applied to confirm the interaction of the newly synthesized compounds with the staphylococcal target proteins associated with biofilm formation [10]. Recently, an antibiofilm algorithm was developed to predict repurposed drugs with potential antibiofilm activity, thus reinforcing the increased interest in the research area under the antivirulence umbrella [11].

In this context, sponge-associated bacteria are known as gifted producers of metabolites with several biologically relevant activities, among them antibiofilm properties [12]. In a previous study, the cell-free supernatant (CFS) obtained from a marine-sponge-associated *Enterobacter* sp. 84.3 culture inhibited biofilm formation (>65%) and dissociated the mature biofilm (>85%) formed by *Staphylococcus aureus* and *Staphylococcus epidermidis* strains. The *Enterobacter* aqueous extract presented concentration-dependent antibiofilm activity for each strain with a minimum biofilm eradication concentration (MBEC) ranging from 16 to 256 μg/mL. In addition, the aqueous extract led to a significant reduction in the mature *S. aureus* biofilm and diminished cell interactions. Importantly, this extract was proven not toxic for mammalian cells (L929 cell line) [13].

Although reports on the antibiofilm potential of *Enterobacter* are somewhat scant, previous studies have demonstrated that this bacterial genus can indeed detain antibiofilm properties, notably owing to the production of *N*-acyl homoserine lactone lactonase [14,15,16,17]. As the sponge-associated *Enterobacter* sp. 84.3 has presented itself as a source of compounds capable of eradicating staphylococcal biofilms with potential applications in the medical field [13], the purpose of this study was to investigate the chemical nature of the active components present in its aqueous extract and the potential dual mode of action of the antibiofilm molecules on staphylococcal cells, unraveling this sponge-derived strain as a fruitful biological resource for the treatment of biofilm-enclosed staphylococci.

## 2. Results

### 2.1. Antibiofilm Activity of the Aqueous Extract

To evaluate the influence of the aqueous extract from *Enterobacter* sp. 84.3 culture supernatant on the staphylococcal biofilm architecture, the mature biofilm (48 h) was analyzed using scanning electron microscopy before and after treatment with the active extract. As shown in Figure 1A,C, the sample prior to treatment was characterized by a thick multilayer biofilm displaying intense interaction among *S. aureus* cells. Interestingly, the treated sample (Figure 1B,D) showed a remarkable reduction in biofilm integrity, evidenced by the loss of the three-dimensional structure and the presence of only a few cellular aggregates. Additionally, it was observed that the mature biofilm of the *S. aureus* ATCC 25923 on the wells of a 96-well microtiter plate was disrupted at 85.3% when treated with 32 μg/mL of the bioactive aqueous extract.

### 2.2. Antibiofilm Molecule Identification

Overall, UHPLC-HRMS revealed the presence of alkaloid, macrolide, steroid, and triterpene molecules in the *Enterobacter* sp. 84.3 bioactive aqueous extract. The detected compounds and their respective chemical classes are presented in Table 1.

### 2.3. Possible Mode of Action on S. aureus Biofilm

A possible dual mechanism of antibiofilm action from the detected metabolites in the aqueous extract of *Enterobacter* sp. 84.3 over the *S. aureus* biofilm is proposed in Figure 2. Briefly, the biofilm matrix can be disrupted by macrolide action, while alkaloids, steroids, triterpenes, and, potentially, macrolide molecules can act by directly inhibiting biofilm formation. Thus, these substances present in the aqueous extract were particularly effective at both hampering biofilm development and disaggregating the mature biofilm, as shown in the scanning electron microscopy images, wherein more than 80% of the biofilm mass was disrupted (Figure 1).

## 3. Discussion

Biofilm formation is a critical factor in the pathogenesis of *S. aureus*, playing a significant role in persistent infections and promoting antimicrobial resistance [18]. The composition of the extracellular matrix varies among strains, ranging from polysaccharide intercellular adhesin (PIA)-dependent to PIA-independent [19,20]. Previous studies have showcased how distinct constituents of the biofilm matrix contribute to its architectural stability and functionality, thus providing the basis for developing novel therapeutics that can effectively target components of the biofilm matrix and modulate biofilm stability [2,3,18,21].

Sponges harbor a diverse array of bacteria that produce a wide range of secondary metabolites, including those with potential antibiofilm activity [22]. Previous results indicate that an *Enterobacter* strain isolated from the marine sponge *Oscarella* sp. displays promising antibiofilm activity against *Staphylococcus* spp. The strain’s aqueous extract inhibited the formation of mature biofilm and induced biofilm disaggregation, without affecting the viability or growth of *Staphylococcus* strains. Importantly, the aqueous extract was also nontoxic for mouse fibroblast cells, thus highlighting an important trait that can be taken into consideration for future applications [13].

In the present study, SEM images further confirmed that the aqueous extract of this sponge-associated bacteria was able to dissociate the mature staphylococcal biofilm. This effect was also observed using confocal scanning laser microscopy, showing a significant reduction in the biofilm layer as well as diminished interactions among the cells [13]. To this extent, the results indicate that the antibiofilm agent could be applied both for the prevention of biofilm formation and the removal of mature biofilm on indwelling medical devices. In addition, seven potential compounds were detected in the aqueous extract of *Enterobacter* sp. 84.3 via LC-HRMS, including alkaloids, macrolides, triterpenes, and steroids. Although it is not possible to affirm via the employed methodology which one of the substances is primarily responsible for the antibiofilm activity, it is highlighted that the aqueous extract is constituted by a complex pool of substances that potentially act synergistically to eradicate the staphylococcal biofilm.

The antibiofilm activity of alkaloids has been previously explored, whereby it was demonstrated that reserpine, an indole alkaloid, inhibits biofilm formation, resulting in the reduced production of extra polymeric substances and reduced biovolume and thickness. Furthermore, docking simulations revealed that the alkaloid had the proteins Bap, IcaA, AtlE, SarA, SarG, and AgrA, which are involved in biofilm composition and regulation, as the most favorable targets [10,23].

Reports on macrolide-mediated antibiofilm activity are somewhat controversial, but there is evidence in vitro and in vivo of such activity on staphylococcal biofilms. For instance, it has been demonstrated that in vitro treatment with clarithromycin and vancomycin was efficient in eliminating staphylococcal biofilms formed on surgical implants [24]. The effects of macrolides in successfully eradicating staphylococcal biofilms were also demonstrated on titanium plates implanted in mice [25]. Microneedles patches impregnated with azithromycin and erythromycin showed antibiofilm effects against *S. aureus* in murine models, resulting in wound healing and skin regeneration [26]. Despite the growing evidence, the subjacent mechanisms of action by which macrolides affect bacterial biofilms remain unelucidated. It has been verified that they might interfere in quorum-sensing (QS) in *S. aureus*, potentially abolishing the biosynthesis of matrix components. Additionally, macrolides could act on mature staphylococci biofilms by likely penetrating the polymeric matrix in a dose- and time-dependent fashion, which has been extensively confirmed in Gram-negative pathogens [27,28].

Although reports on steroid-producing bacteria are scant and mostly restricted to *Myxobacteria* spp. [29], here, the presence of two substances was indicated, and they were classified as steroids. Among these, a fluoride steroid was detected, which is a common trait in marine bacterial metabolites, including those with antibiofilm activity [30]. In a previous study, the biofilm biomass of *S. aureus* was significantly reduced in the presence of steroids [31]. Steroids derivatives have had their antibiofilm potential recently described, whose action was accompanied by the downregulation of the *fnbB* (fibronectin-binding protein B), *capC* (capsule biosynthesis protein C) genes, and the *isaA* gene (immunodominant staphylococcal antigen) [32]. Proteome analyses of *S. aureus* biofilm have already indicated that proteins associated with adhesion, peptidoglycan synthesis, and fibrinogen binding are most predominant, whereas virulence-related proteins, such as IsaA, are expressed in lower amounts [33]. Interestingly, a previous study has shown that treatment with steroid derivatives led to the upregulation of *isaA*, and this could be implicated in the reduction in staphylococcal biofilm formation [32].

Triterpenes have also been reported as promising antibiofilm agents. Their activity has been previously associated with alterations in the structure of bacteria, such as the cell membrane and adhesins, and may also interfere with gene expression, thus leading to a lower capacity of adhesion and, consequently, biofilm formation [34]. For instance, ursolic acid at concentrations of 10 and 30 μg/mL has been associated with the induction of several genes related to chemotaxis and mobility, particularly the gene *motAB*, leading to the excessive motility of strains of *Escherichia coli* and preventing the establishment of the bacterial biofilm [35]. Further, the effect of triterpene derivatives has been assessed on *S. aureus* biofilm formation, whereby not only significant biofilm inhibition was observed, but also no antibiotic or cytotoxic effects were observed [36].

Based on the results from the chemical characterization of the aqueous extract, we suggest a potential dual mechanism of antibiofilm action by the metabolites produced by the *Oscarella*-derived *Enterobacter* sp. Regardless of their chemical classes, it is likely that all metabolites may inhibit biofilm formation by affecting the production of matrix components, as suggested by the previous abovementioned reports of the antibiofilm properties of the compounds identified [24,25,31,36]. This can take place by either a direct interaction of these metabolites with structural elements of the biofilm matrix, such as the aforementioned for alkaloids [23], or by hindering the gene regulation of biofilm formation, which could be via QS interference, as pointed out for the macrolides [28], or an imbalance at the gene expression underlying the biofilm structuring, as highlighted for the steroids [31]. Moreover, the detected macrolide is potentially the best choice among the metabolites present in the *Enterobacter* extract capable of dissociating the mature staphylococcal biofilm [24]. Hence, this metabolite mix functions synergistically in the efficient removal of *Staphylococcus* spp. biofilm in all potential stages of its life cycle. Importantly, this extract did not kill or inhibit bacterial growth, thus suggesting its antivirulence properties [13]. Therefore, following the treatment with the sponge bacterial extract, staphylococci cells are free from biofilm protection and susceptible to antimicrobial chemotherapy and immune response.

## 4. Materials and Methods

### 4.1. Bacterial Strains and Culture Conditions

Sponge-associated *Enterobacter* sp. 84.3 (MK780772.1) was isolated from the Brazilian sponge *Oscarella* spp. as previously described [13]. Cell-free culture supernatants were prepared by inoculating 10^7^ cells of *Enterobacter* 84.3 in 3.0 mL of BHI medium (Difco, Tucker, GA, USA). Following incubation at 25 °C for 24 h, 22 mL of BHI was added, and the culture was further incubated at 25 °C for 48 h. After centrifugation at 4000× *g* for 20 min, the CFS was sterilized using filtration (Millipore 0.22 μm) and used for antibiofilm activity assays as previously described [13].

The reference strain of the American Type Culture Collection (ATCC) *S. aureus* ATCC 25923, classified as a strong biofilm producer, was included in this study as an indicator strain and positive control for further antibiofilm assays. It was grown on BHI agar or in BHI broth supplemented with 1% glucose (BHIg) for 18 h at 37 °C.

### 4.2. Fermentation Conditions and Extract Preparation

The bioactive extract, previously identified as the delipidated aqueous phase, was obtained as previously described [13]. Briefly, total biomass obtained from a 500 mL culture in BHI was submitted twice to fractionation with 200 mL *n*-hexane (Tedia, Fairfield, OH, USA). The hexane solution was then removed under vacuum, and the remaining aqueous residue was extracted twice (200 mL) using the following solvents in this order: dichloromethane, ethyl acetate, and butanol (Tedia, USA). The organic fractions were evaporated and tested for antibiofilm activities. The delipidated aqueous phase was filtered (Millipore 0.22 μm), tested, and named as the aqueous extract.

### 4.3. Scanning Electron Microscopy Observation of Antibiofilm Activity

To observe the multicellular structures in the biofilm in the presence or absence of the bioactive extract, aliquots of overnight culture (~10^8^ CFU/mL) of *S. aureus* ATCC 25923 were distributed on chamber slides (Nunc Inc., Rochester, NY, USA) for scanning electron microscopy (SEM) observations. After incubation at 37 °C for 48 h, bacteria that remained in suspension were removed by aspiration, and the remaining adherent cells on the slide were washed three times with PBS and treated with the bioactive substance at the previously established minimum biofilm eradication concentration (MBEC) of 32 μg/mL [13]. An untreated biofilm control was included on each slide. The slides were again washed with PBS and fixed in 2.5% glutaraldehyde in 0.2 M sodium cacodylate buffer (pH 7.4) at 25 °C. Then, washing was carried out thrice in 10 mL sodium cacodylate buffer 0.2 M solution, followed by dehydration in a graded ethanol series at 30%, 50%, 70%, 90%, and 100% (15 min each and 3× at the last concentration). Drying at the critical point was carried out with carbon dioxide. At the metallization step, samples were sputter-coated with gold (~5 nm layer). Biofilms on chamber slides were analyzed using a Quanta 250 Microscope (FEI Company, Hillsboro, OR, USA) in high vacuum SEM mode.

To evaluate the antibiofilm effect of the aqueous extract (32 μg/mL), the same strategies described for SEM analysis were carried out in the wells of a 96-well microtiter plate, as previously described [13]. Briefly, in the wells of a 96-well microtiter plate (TPP, Trasadingen, Switzerland), 20 μL of the indicator strain culture (~10^8^ CFU/mL) was mixed with 180 μL of BHIg, and the plate was incubated aerobically at 37 °C for 24 h. After incubation, planktonic cells were removed by washing the plate three times with sterile phosphate-buffered saline (PBS; pH 7.4). To the remaining preformed staphylococcal biofilm, 200 μL of the CFS was added per well. Control samples of staphylococcal biofilm were kept without the addition of supernatant. After a second incubation at 37 °C for 24 h, the plates were again washed with PBS and heat-fixed at 60 °C for 60 min. The wells were then stained with 0.2% crystal violet solution for 15 min at 25 °C. The plate was air-dried, and the dye retained by adherent cells was dissolved in 95% ethanol. The optical density of each well containing the mature biofilms treated and untreated after crystal violet staining was measured at 570 nm using a microtiter plate reader (model 680, Bio-Rad Laboratories, Watford, UK). The ability to disrupt biofilm of indicator strain was expressed as a percentage of antibiofilm activity by applying the following formula: (ODcontrol − ODsample/ODcontrol) × 100. Three independent experiments were performed in triplicate; therefore, each data point was averaged from a total of nine values, and the standard deviation (SD) was calculated.

### 4.4. Chemical Analysis, Bioinformatics Screening, and Compound Identification

#### 4.4.1. UHPLC-MS Analysis of the Aqueous Extract

To identify the bioactive molecule(s) in the aqueous extract derived from *Enterobacter* sp. 84.3, ultra-high-performance liquid chromatography coupled with high-resolution mass spectrometry (UHPLC-HRMS) was carried out. The analyses were performed in a modular Nexera UHPLC (Shimadzu, Kyoto, Japan) system with LC-30AD pump, SIL-30AC autosampler, CTO-30A column oven, and SPD-M20A DAD detector (spectra were recorded from 200 to 800 nm). The LC-system was connected to a QTOF-HRMS mass spectrometer (Bruker, Billerica, MA, USA) fitted with an ESI source (negative mode). Negative ion mass spectra were recorded in the range of *m*/*z* 50–1.500 at a scan speed of 1.5 kDa/s at a voltage of −5 kV and capillary temperature of 280 °C. Nitrogen was used as the gas source. The components were separated using a Shimpack XR II series C18 (120 mm × 3.6 mm × 2.1 μm). Samples were solubilized in methanol (MeOH) MS-LC grade solvent (3 mg/mL), and the injection volume was 3 μL for the samples. The mobile phases used were 0.1% (*v*/*v*) formic acid in water (phase A) and acetonitrile (phase B) at a flow rate of 0.3 mL/min. The gradient program was as follows: 3% B (0.01 min), 3–15% B (10 min), 15–45% B (12 min), 45–100% B (4.5 min), 100% B (2.5 min), and 100–0% B (4 min).

#### 4.4.2. MZmine Parameters

Raw files were converted to universal mzXML files using MSConvert, part of the ProteoWizard package [37] and directly processed in MZmine 2.1 [38,39]. Mass detection was performed keeping the noise level at 1. Chromatogram building was achieved using a minimum time span of 0.3 min, a minimum height of 1.0 × 10^2^, and *m*/*z* tolerance of 0.002 (or 5 ppm). The local minimum search deconvolution algorithm was used with the following settings: chromatographic threshold = 5.0%, minimum retention time range = 0.2 min, minimum relative height = 15.0%, minimum absolute height = 5.0 × 10^5^, minimum ratio of peak top/edge = 5, and peak duration range = 0.3–10.0 min. Chromatograms were deisotoped using the isotopic peaks grouper algorithm with a *m*/*z* tolerance of 0.002 (or 5 ppm) and a RT tolerance of 0.2 min. Peak alignment was performed using the Join aligner method with a *m*/*z* tolerance at 0.002 (or 5 ppm), weight for *m*/*z* = 15, RT tolerance = 0.2 min, weight for RT = 10, minimum absolute intensity = 5.0 × 10^5^, and minimum score = 65%. The peak list was eventually gap-filled with the peak finder module (intensity tolerance at 1.0%, *m*/*z* tolerance at 0.002 (or 5 ppm), and absolute RT tolerance of 0.2 min). Compound identification was performed using online searches against the KEGG Database release 80.

## 5. Conclusions 

In conclusion, this report supports the potential of sponge-derived *Enterobacter* as a valuable source of antibiofilm compounds. The aqueous extract could inhibit the biofilm formation and dissociate the mature biofilm of *S. aureus*, and seven potential compounds were proposed, including alkaloids, macrolides, steroids, and triterpenes. These results highlight promising strategies, including multitargeted ones, which could also be applied with combinational therapies to eradicate staphylococcal biofilms. The study not only presents the novelty in exploring the antibiofilm properties of *Enterobacter* spp. but also suggests a dual mode of action for the antibiofilm compounds.

## Figures and Tables

**Figure 1 molecules-28-04843-f001:**
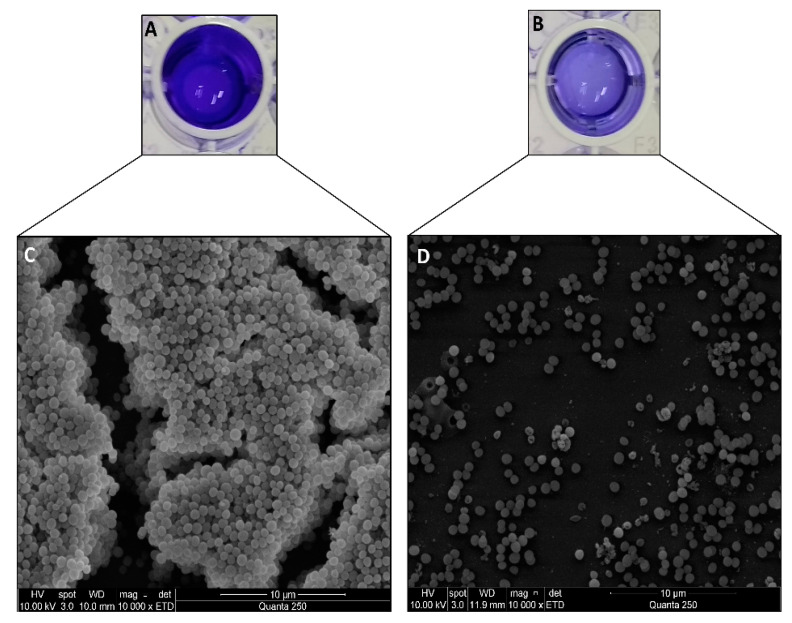
Representative scanning electron microscopy images of *S. aureus* ATCC 25923 biofilm in the presence of the *Enterobacter* sp. 84.3 bioactive aqueous extract (32 μg/mL). (**A**,**B**) Wells of 96-well plate showing mature biofilm untreated (left well) and treated (right well) after crystal violet staining. Loss of color intensity indicates biofilm disaggregation. (**C**) Untreated biofilm control and (**D**) treated biofilm. The images show the best antibiofilm effect compared to the control.

**Figure 2 molecules-28-04843-f002:**
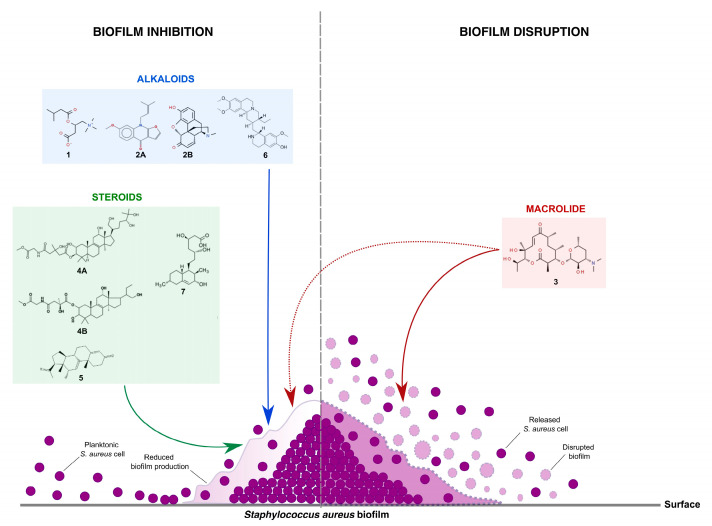
Proposed dual mechanism of antibiofilm action of detected metabolites in the aqueous extract of *Enterobacter* sp. 84.3 over *S. aureus* biofilm (center of the image, colored and shaded in purple). While the macrolide (**3**. novamethymycin; shaded in red) can potentially act by both disrupting the biofilm matrix (red arrow) and inhibiting its formation (dashed red arrow), the alkaloids (**1**. isovalerylcarnitine; **2**. acrophylline (**A**) or morphinone (**B**); **6**. cephaeline, shaded in blue), triterpenes (**4**. fasciculol E (**A**) or F (**B**), shaded in green), and steroids (**5**. 12-methylpregna-4,9(11)-diene-3,20-dione, (12alpha)-; **7**. (3alpha,5beta,11beta,17beta)-9-fluoro-17-methylandrostane-3,11,17-triol; shaded in gray) likely interfere in biofilm production (blue, green, and gray arrows, respectively). Cells of *S. aureus* (represented by purple circles) are eventually released as the biofilm is dismantled by the synergistic action of these metabolites.

**Table 1 molecules-28-04843-t001:** Compounds putatively identified through KEGG Library from sponge-associated *Enterobacter* sp. aqueous extract analyzed using UHPLC-HRMS.

Compounds	Name	Class	RT (min)	Area	Molecular Ion [M − H]^−^
**1**	Isovalerylcarnitine	Alkaloid	1.01	5.0 × 10^5^	244.1554
**2**	Acrophylline or morphinone	Alkaloid	1.42	1.2 × 10^6^	282.1125
**3**	Novamethymycin	Macrolide	2.74	3.3 × 10^5^	484.2883
**4**	Fasciculol E or F	Tetracyclic triterpene	11.37	3.2 × 10^5^	722.4543
**5**	12-methylpregna-4,9(11)-diene-3,20-dione	Steroid	13.51	3.1 × 10^5^	325.2181
**6**	cephaeline	Alkaloid	14.73	4.6 × 10^5^	465.2727
**7**	(3alpha,5beta,11beta,17beta)-9-fluoro-17-methylandrostane-3,11,17-triol	Steroid	15.54	5.7 × 10^5^	339.2165

## Data Availability

Not applicable.

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
