# Peer review of "Delving into the Mechanisms of Sponge-Associated Enterobacter against Staphylococcal Biofilms"

_molecules, 2023, doi:10.3390/molecules28124843_

Round 1
Reviewer 1 Report
The manuscript “Staphylococcal biofilm impairment by sponge-associated Enterobacter: delving into the mechanisms behind its antibiofilm activity” is a very interesting work. Staphylococci are one of the most common causes of biofilm-related infections. Such infections are hard to treat with conventional antimicrobials, which often lead to bacterial resistance, thus being associated with higher mortality rates while imposing a heavy economic burden on the healthcare system. Investigating antibiofilm strategies is an area of interest in the fight against biofilm-associated infections. Previously, the cell-free supernatant from marine sponge-associated Enterobacter sp. inhibited staphylococcal biofilm formation and dissociated the mature biofilm. This study aimed to identify the chemical components responsible for the antibiofilm activity of Enterobacter sp. Scanning electron microscopy confirmed that the aqueous extract at the concentration of 32 µg/mL could dissociate the mature biofilm. Liquid chromatography coupled with high-resolution mass spectrometry revealed seven potential compounds in the aqueous extract, including alkaloids, macrolides, steroids, and triterpenes. While I believe this topic is of great interest to our readers, I think it needs major revision before it is ready for publication. So, I recommend this manuscript for publication with major revisions.
1. In this manuscript, the authors did not explain the importance of antibiofilm activity in the introduction part. The authors should explain the importance of antibiofilm activity.
2) Title: The title of the manuscript is not impressive. It should be modified or rewritten it.
3) Correct the following statement “This study also discussed the putative mode of action on staphylococcal cells and supports the potential of sponge-derived Enterobacter as a source of antibiofilm compounds”.
4) Keywords: There are so many keywords and reduce them up to 5. So, modify the keywords.
5) Introduction part is not impressive. The references cited are very old. So, Improve it with some latest literature like 10.3390/molecules27196457, 10.3390/molecules27217368.
6) The authors should explain the following statement with recent references, “It is primarily protein-dependent, but the composition of the extracellular matrix varies among strains”.
7) Add space between magnitude and unit. For example, in synthesis “21.96g” should be 21.96 g. Make the corrections throughout the manuscript regarding values and units.
8) The author should provide reason about this statement “Regardless of their chemical classes, it is likely that all metabolites inhibit biofilm formation by affecting the production of matrix components”.
9) Comparison of the present results with other similar findings in the literature should be discussed in more detail. This is necessary in order to place this work together with other work in the field and to give more credibility to the present results.
10) Conclusion part is very long. Make it brief and improve by adding the results of your studies.
11) There are many grammatic mistakes. Improve the English grammar of the manuscript.
Minor editing of English language required
Author Response
Dear,
Thank you very much for reviewing our manuscript and for reconsidering it for publication. The present version has been revised and all corrections have been incorporated. The changes have been red highlighted in the text.
Reviewer #1
1) In this manuscript, the authors did not explain the importance of antibiofilm activity in the introduction part. The authors should explain the importance of antibiofilm activity.
In the first and second paragraphs, we have mentioned the consequences of biofilm-associated infections and have explained that antibiofilm strategies could play a pivotal role in their treatment.
2) Title: The title of the manuscript is not impressive. It should be modified or rewritten it.
The title has been rewritten.
3) Correct the following statement “This study also discussed the putative mode of action on staphylococcal cells and supports the potential of sponge-derived Enterobacter as a source of antibiofilm compounds”.
This statement has been corrected.
4) Keywords: There are so many keywords and reduce them up to 5. So, modify the keywords.
The keywords have been revised.
5) Introduction part is not impressive. The references cited are very old. So, Improve it with some latest literature like 10.3390/molecules27196457, 10.3390/molecules27217368.
We thank the reviewer for the suggestion. To fulfill the reviewer’s remark, we have included more recent references in the introduction.
6) The authors should explain the following statement with recent references, “It is primarily protein-dependent, but the composition of the extracellular matrix varies among strains”.
This sentence has been rewritten for clarity purposes.
7) Add space between magnitude and unit. For example, in synthesis “21.96g” should be 21.96 g. Make the corrections throughout the manuscript regarding values and units.
This issue has been revised.
8) The author should provide reason about this statement “Regardless of their chemical classes, it is likely that all metabolites inhibit biofilm formation by affecting the production of matrix components”.
This sentence has been revised and a reason was proposed based on the field literature.
9) Comparison of the present results with other similar findings in the literature should be discussed in more detail. This is necessary in order to place this work together with other work in the field and to give more credibility to the present results.
We agree, similar findings in the literature have been presented in order to contextualize the derived results with what has been currently published in the field.
10) Conclusion part is very long. Make it brief and improve by adding the results of your studies.
The conclusion has been revised according to the reviewer’s suggestions.
11) There are many grammatic mistakes. Improve the English grammar of the manuscript.
The English grammar has been improved.
Should any question arise, please do not hesitate to contact me. Thank you very much for your consideration.
Best wishes,
Reviewer 2 Report
- The research carried out by the Canellas et al. focuses on characterizing the anti-biofilm activity of Sponge-associated Enterobacter against Staphylococcal biofilms. I have some minor issues regarding the manuscript:
- Include some relevant literature, e.g., doi.org/10.3389/fmars.2022.980418 and doi.org/10.1093/nar/gkx1157 (aBiofilm database), etc., in the 'Introduction' or 'Discussion' section
- I want to ask the authors to calculate the IC50 value. The current version of the manuscript includes the disruption of 85.3% when treated with 32 µg/mL.
- The ‘mode of action’ described by the authors looks exaggerated. To establish the ‘mode of action,’ the author should include the validation strategies like molecular modeling or try to find any relevant literature suggesting the same. Try to elaborate the ‘Result’ section of the ‘Mode of Action’ with relevant literature.
English editing is required
Author Response
Dear,
Thank you very much for reviewing our manuscript and for reconsidering it for publication. The present version has been revised and all corrections have been incorporated. The changes have been red highlighted in the text.
Reviewer #2
1) Include some relevant literature, e.g., doi.org/10.3389/fmars.2022.980418 and doi.org/10.1093/nar/gkx1157 (aBiofilm database), etc., in the 'Introduction' or 'Discussion' section
We thank the reviewer for the suggestion. We have included more recent references in both sections in this revised version of the manuscript.
2) I want to ask the authors to calculate the IC50 value. The current version of the manuscript includes the disruption of 85.3% when treated with 32 µg/mL.
The IC50 value was not calculated, however, the antibiofilm activity of the aqueous extract from Enterobacter sp. 84.3 culture on staphylococcal biofilm was analysed as previously reported (doi.org/10.1016/j.resmic.2020.10.002) by biofilm reduction rate by applying the following formula: (ODcontrol - ODsample/ODcontrol) x 100. The ability to dissociate biofilm was also expressed as antibiofilm activity (%), by applying the above-mentioned formula and the minimum biofilm eradication concentration (MBEC) was calculated in the aforementioned preeceding work and mentioned in the current manuscript.
3) The ‘mode of action’ described by the authors looks exaggerated. To establish the ‘mode of action,’ the author should include the validation strategies like molecular modeling or try to find any relevant literature suggesting the same. Try to elaborate the ‘Result’ section of the ‘Mode of Action’ with relevant literature.
In our study, we have made an effort to thoroughly discuss our findings in the context of the existing literature encompassing antibiofilm compounds of bacterial origin, which is still scarce, particularly taking into account the chemical classes of the unveiled metabolites. We have conducted a review of the available research and have incorporated relevant studies into our discussion. Notably, more recent references have been included in the discussion section in this revised version of the manuscript. Additionally, we have made an effort to lower the tone regarding the proposed dual mechanism of action of the detected molecules in the aqueous extract from the sponge-derived Enterobacter, which remains as a proposal considering what has been currently published in the field
Should any question arise, please do not hesitate to contact me. Thank you very much for your consideration.
Best wishes
Reviewer 3 Report
The authors reported Staphylococcal biofilm impairment by sponge-associated Enterobacter: delving into the mechanisms behind its antibiofilm activity.
There are following major comments as follows:
1. The introduction is needed to be improved. Therefore, it is recommended to include follows references: Chitosan-gum arabic embedded alizarin nanocarriers inhibit biofilm formation of multispecies microorganisms, Carb. poly. 2022
2. the rationale of research must be demonstrated graphically in the last section of introduction.
3. If possible, the more identification and characterization can be performed to confirm the extract ingredients such as NMR, Mass etc
4. For the biofilm confirmation, it is recommended to be included the confocal and SEM analysis.
Author Response
Dear,
Thank you very much for reviewing our manuscript and for reconsidering it for publication. The present version has been revised and all corrections have been incorporated. The changes have been red highlighted in the text.
Reviewer #3
1) The introduction is needed to be improved. Therefore, it is recommended to include follows references: Chitosan-gum arabic embedded alizarin nanocarriers inhibit biofilm formation of multispecies microorganisms, Carb. poly. 2022
We thank the reviewer for the suggestion. We have included more recent references in the introduction.
2) The rationale of research must be demonstrated graphically in the last section of introduction.
The rationale of research is graphically demonstrated in the form of a graphical abstract of this manuscript.
3) If possible, the more identification and characterization can be performed to confirm the extract ingredients such as NMR, Mass etc
We carefully considered your comments and understand your interest in further analysis. However, we are currently unable to conduct the requested analysis due to limitations beyond the scope of this study. Nonetheless, we have provided further explanations where necessary and clarified our findings based on the available data.
4) For the biofilm confirmation, it is recommended to be included the confocal and SEM analysis.
In the present study, SEM analysis have been performed for observation of antibiofilm activity, please, see the Figure 1. In a previous study, the effect of the same aqueous extract on mature S. aureus biofilm was also observed by confocal scanning laser microscopy. It showed a significant reduction of the biofilm layer as well as diminished interactions among the cells (doi.org/10.1016/j.resmic.2020.10.002).
Should any question arise, please do not hesitate to contact me. Thank you very much for your consideration.
Best wishes
Round 2
Reviewer 3 Report
The revised manuscript can be proceeded in journal for further process.